



**Clay mineralogical evidence for mid-latitude terrestrial climate change from the**
**latest Cretaceous through the earliest Paleogene in the Songliao Basin, NE China**
Yuan Gao[1*], Youfeng Gao[2], Daniel E. Ibarra[3,4], Xiaojing Du[5], Tian Dong[1], Zhifei Liu[6], Chengshan
Wang[1]
*1 State Key Laboratory of Biogeology and Environmental Geology, School of Earth Sciences and Resources, China*
*University of Geosciences (Beijing), Beijing 100083, China*
*2 Key Lab for the Evolution of Past Life and Environment in Northeast Asia, Ministry of Education, Jilin University,*
*Changchun 130026, China*
*3 Department of Earth and Planetary Science, University of California, Berkeley, California 94720 USA*
*4 Institute at Brown for Environment and Society and the Department of Earth, Environmental and Planetary Science,*
*Brown University, Providence, Rhode Island 02912 USA*
*5 Department of Earth and Environmental Science, University of Michigan, Ann Arbor, MI 48109, USA*
*6 State Key Laboratory of Marine Geology, Tongji University, Shanghai 200092, China*
*Correspondence to: Yuan Gao (yuangao@cugb.edu.cn)*
**Abstract**
From the latest Cretaceous (late Campanian to Maastrichtian, ~76-66 Ma) through the earliest
Paleogene, a fluctuating greenhouse climate prevailed and climatic changes were linked to
catastrophic geological events and massive biotic extinction. Paleoclimate reconstructions during this
time period primarily rely on marine sediments, with limited high-resolution terrestrial records. Here
we present a high-resolution clay mineralogical record from the Sifangtai Formation and the Mingshui
Formation of the Songliao Basin, northeast China, which are continuously deposited fluvial to
lacustrine strata, and have been tightly age constrained as late Campanian to early Danian. Smectite





and illite are the dominant clay species, whereas kaolinite and chlorite are minor components. Clay
minerals are derived from the weathering of parent rocks and/or paleosols, and their relative weight
percentages are primarily controlled by regional paleoclimate and sedimentary environment. We use
three clay mineralogical proxies, including the percentage ratio of smectite and illite, illite chemistry
index and the percentage ratio of phyllosilicate clay minerals and quartz in clay fractions, for
paleoclimatic reconstruction. We correlate these proxy timeseries with basin-scale and global
paleoclimate timeseries. Our results show that from the latest Cretaceous through the earliest
Paleogene, values of all three clay mineralogical proxies in the Songliao Basin are generally higher
during warming intervals than those during cooling intervals. We interpret this dataset to suggest that
warming caused strengthened moisture delivery from the Pacific, increasing precipitation and
intensified chemical weathering, whereas cooling was accompanied by increasing dryness and
physical weathering. Before the Cretaceous-Paleogene (K-Pg) boundary (approximately 66.4 Ma to
66.0 Ma), the warming likely related to Deccan volcanism and the transient cooling afterwards are
characterized by paleosol carbonate stable isotopic excursions and changes in the illite chemistry
index recorded in the Songliao Basin sediments, reflecting fluctuations in precipitation and weathering
intensity. However, changes in clay mineral assemblages are not clear before and at the K-Pg
boundary. This is probably due to the relatively long-response time of terrestrial weathering regimes
(up to 500 kyrs) to the short duration of the K-Pg boundary impact and the degassing by the preceding
Deccan Traps volcanism (~200 kyrs). In the earliest Paleogene, after the K-Pg boundary, all clay
mineralogical and stable isotopic proxies indicate a warmer and more humid climate with stronger
chemical weathering. Our work demonstrates that terrestrial climate and weathering intensity in the
mid-latitude Songliao Basin fluctuated during the latest Cretaceous through the earliest Paleogene and
sensitively responded to global climate changes.
**Keywords:** clay mineral, terrestrial paleoclimate, weathering, Cretaceous-Paleogene Boundary,
Songliao Basin



## 1. Introduction

The Late Cretaceous witnessed a global cooling trend following peak warmth during the mid-Cretaceous, an interval of the warmest greenhouse climate during the past 100 million years (Friedrich et al., 2012; Linnert et al., 2014; O'Brien et al., 2017). From the latest Cretaceous (late Campanian to Maastrichtian, ~76-66 Ma) through the earliest Paleogene, short-term climatic oscillations were superimposed on the long-term trend of decreasing temperatures (Berrera and Savin, 1999; Friedrich et al., 2012; Gao et al., 2015a). Evidence for short-term climate cooling, most prominent in the early Maastrichtian, are inferred primarily from positive $\delta^{18}O$ values in benthic foraminifera in multiple ocean basins, and sedimentary and palynological records suggesting the presence of sea ice in polar regions (Berrera and Savin, 1999; Davies et al., 2009; Friedrich et al., 2012; Bowman et al., 2013). Warming events at hundred-thousand to million-year time scales occurred during the middle Maastrichtian, the latest Maastrichtian and the earliest Paleocene, which are supported by records of the oxygen and clumped isotopes of biogenic carbonates and records of the organic geochemical proxy TEX$_{86}$ for sea surface temperatures (e.g., Friedrich et al., 2012; Petersen et al., 2016; Woelders et al., 2018; Hull et al., 2020). Understanding the processes and mechanisms of these climatic oscillations are essential for deciphering their cause-and-effect relationships with catastrophic events of the latest Cretaceous, such as the Deccan Traps volcanism and the Chicxulub asteroid impact (Alvarez et al., 1980; Keller et al., 2012; Schoene et al., 2019; Sprain et al., 2019; Hull et al., 2020). Furthermore, it has been debated whether these climatic oscillations are tightly linked to biological evolution from the latest Cretaceous through the earliest Paleogene, including at the Cretaceous-Paleogene (K-Pg) boundary mass extinction, during the pre-K-Pg-boundary high biologic stress and during the post-K-Pg-boundary biotic recovery (Keller et al., 2012; Petersen et al., 2016; Lyson et al., 2019).

Most of the climatic and biological records from the latest Cretaceous through the earliest Paleogene are derived from marine sediments. Continuous terrestrial records during this time interval that have high-resolution age constraints and multiple paleoenvironmental proxies comparable to



marine records are still lacking, despite efforts on terrestrial studies during the past two decades
(Nordt et al., 2003; Wilf et al., 2003; Tobin et al., 2014; Gao et al., 2015a; Sprain et al., 2015). The
Songliao Basin in northeastern China, which was a long-lived lake basin in the Cretaceous Period,
preserved up to 10,000 meters of terrestrial sediments (Feng et al., 2010; Wang et al., 2013; Figure
1A). An International Continental Scientific Drilling Project in the Songliao Basin, the SK-1 scientific
boreholes, recovered 800-meters of nearly complete and continuous cores of terrestrial strata spanning
the latest Cretaceous to the earliest Paleogene, namely the Sifangtai Formation and the Mingshui
Formation (Feng et al., 2013; Wang et al., 2013; Gao et al., 2019). These two geological formations
have been precisely age constrained as 76.1 to 65.1 Ma by magnetostratigraphy, biostratigraphy and
cyclostratigraphy (Deng et al., 2013; Wan et al., 2013; Wu et al., 2014). Previous studies using stable
isotopes of fossil ostracods and pedogenic carbonates from the Sifangtai and Mingshui Formations
(SMF) of the SK-1 cores demonstrate a punctuated terrestrial climate in response to global climatic
oscillations (Chamberlain et al., 2013; Gao et al., 2015a; Zhang et al., 2018). However, further
paleoclimatic and paleoenvironmental proxies are needed to illustrate and elucidate the terrestrial
climatic evolution of the Songliao Basin, in particular changes in hydroclimate and weathering
intensity, from the latest Cretaceous through the earliest Paleogene.

Sedimentary clay minerals are phyllosilicates finer than 2 μm that are commonly formed in

weathering profiles and soils (Chamley, 1989; Thiry, 2000; Liu et al., 2012). These minerals are
useful indicators for paleoenvironmental and paleoclimatic evolutions in modern and deep-time
sedimentary basins, on the basis that post-depositional processes do not significantly alter the
mineralogical composition (Chamley, 1989; Thiry, 2000; Sáez et al., 2003; Dera et al., 2009; Liu et al.,
2010; Gao et al., 2018; Deconinck et al., 2019). Efforts have been made to study the clay mineral
assemblages through the SMF in the Songliao Basin and specifically on the SK-1 cores (Gao et al.,
2013; 2015b). Gao et al. (2015b) studied clay mineral compositions throughout both of the  ~2500 m
long SK-1 cores at a sampling interval of ~10 m and interpreted paleoenvironmental and post-
deposition signals. It was demonstrated that in the latest Cretaceous SMF, authigenic kaolinite and
smectite present in sandstones whereas were formed during early diagenesis whereas clay mineral





compositions in mudstones were primarily controlled by paleoenvironmental factors. Further, a more
detailed study of clay mineralogy of a 60 m thick section of the middle Mingshui Formation in SK-1
cores indicates that variations in smectite and illite composition mainly reflect changes on regional
paleoclimate and provenance (Gao et al., 2013). In this paper, we present a high-resolution (~1-5 m
sampling interval) clay mineralogical sequence throughout SMF in the SK-1 cores, based on 213 new
and 91 published data (Gao et al., 2013; 2015b). By correlating clay mineral variations to regional and
global paleoclimate timeseries, we interpret the terrestrial paleoclimatic evolution in the Songliao
Basin and link these changes to global climate from the latest Cretaceous through the earliest
Paleogene.

**2. Geological setting**
The Songliao Basin lies in northeastern margin of the Eurasian continent and covers roughly
260,000 km$^2$ in the northeastern China (Fig. 1; Feng et al., 2010; Wang et al., 2013). It was formed as
a rift basin in the late Mesozoic extensional domain of eastern China and eastern Mongolia, as a result
of interactions among the Pacific plate, the North China craton and the Siberia craton (Graham et al.,
2001; Feng et al., 2010; Wang et al., 2016). During the Cretaceous, the Songliao Basin underwent
three distinct tectonic episodes, including rifting, thermal subsidence and structural inversion (Feng et
al., 2010; Wang et al., 2016). In the Early Cretaceous rifting stage, multiple fault blocks developed
along NNE trending and volcaniclastic and sedimentary rocks deposited in isolated basins (Feng et al.,
2010; Wang et al., 2016). Continued extension and lithospheric cooling caused regional thermal
subsidence from Early Cretaceous through Late Cretaceous, during this time thousands-of-meters of
alluvial fan, fluvial and lacustrine sediments were preserved in the basin (Feng et al., 2010; Wang et
al., 2013). In the late stage of the basin evolution, subduction of the Pacific Plate beneath the Eurasian
continent caused significant regional compression and basin-scale structural inversion, therefore the
sedimentary basin shrunk to demise (Wang et al., 2016; Zhang et al., 2017). The "Continental
Scientific Drilling Project of Cretaceous Songliao Basin (SK)" is aimed to obtain a complete,





continuous, terrestrial sedimentary record of the whole Cretaceous and to study continental climate
and environment during the greenhouse period (Feng et al., 2013; Wang et al., 2013; Gao et al., 2019).
In its first phase, the SK-1 project, two scientific boreholes (SK-1s and SK-1n) have been conducted
to recover rock cores of 2,485.89 m in total length with a recovery ratio of 96.46% (Feng et al., 2013;
Gao et al., 2019).

The Sifangtai and Mingshui Formations (SMF) are the uppermost Cretaceous units in the

Songliao Basin deposited during the structural inversion stage of basin development (Feng et al., 2010;
Wang et al., 2013). Two regional unconformities, which represent hiatus spanning millions of years as
a result of intensified tectonic compression, separate these two formations from underlying and
overlying strata, although no obvious unconformities have been detected within the SMF stratigraphy
including between the Sifangtai and Mingshui Formations (Feng et al., 2010). The SMF is mainly
comprised of grey-green and brown-red colored mudstones, siltstones and fine sandstones, deposited
under fluvial to shallow lacustrine environments (Feng et al., 2010; Wang et al., 2015). In the SK-1n
scientific core, the Sifangtai Formation (depth range of 807.12-1021.60 m) is characterized by reddish
to brownish mudstone, siltstone and grayish sandstone of fluvial to environments (Figure 2; Wang et
al., 2015). The Mingshui Formation (depth range of 807.12-210.66 m) is subdivided into two
members according to lithology and sedimentary facies. The lower member, the lower ~200 m of the
formation, is characterized by grey siltstone, sandstone and two sets of grey to black mudstone with
fine laminations (Figure 2; Wang et al., 2015). These mudstones were interpreted as shallow to semi-
deep lacustrine facies, probably controlled by temporarily intensified extensional stress field (Cheng et
al., 2009; Zhang et al., 2009; Wang et al., 2015). The upper ~400 m of the Mingshui Formation is
characterized by green, brown and red mudstone and grey siltstone and sandstone, mainly deposited
under fluvial and shore to shallow lacustrine environments (Figure 2; Cheng et al., 2009; Wang et al.,
2015). Paleosols have been identified in the floodplain mudstones and shore lacustrine mudstones
throughout the SMF (Huang et al., 2013; Gao et al., 2015a).





The ages of the SMF sediments in the SK-1n core are well constrained by multiple

geochronological efforts. Paleomagnetic studies indicate that the SMF spans five magnetozones from
C33n to C29r in the Geomagnetic Polarity Time Scale (Deng et al., 2013). This is consistent with
biostratigraphic studies on ostracods, pollens and spores, and charophytes, which suggest a late
Campanian to early Danian age (Li et al., 2011; Wan et al., 2013; Qu et al., 2014; Li et al., 2019).
Obvious decameter-to meter-scale sedimentary cycles in thorium logging data in SMF reflect
Milankovith cycles, allowing for the establishment of a robust astronomical time scale by tuning
filtered 405 kyr eccentricity cycles (Wu et al., 2014). Anchored by the boundary of paleomagnetic
chrons C29r/C30n at 342.1 m with an absolute age of 66.3 Ma, this astronomical time scale provides
precise age control for the SMF sediments, which is applied for all samples in this study. Furthermore,
the astronomical time scale places the Cretaceous-Paleogene Boundary (66.0 Ma) at a depth of ~318
m, which is in agreement with constraints generated from other chronological methods (Wan et al.,
2013; Wu et al., 2014).

**3. Methods**

In this study 91 published and 213 new data were used for clay mineralogical analysis in the

SMF of the SK-1n core. In general data points are evenly distributed throughout the mudstones in the
SMF with a depth resolution of approximately 1 m to 5 m, corresponding to a temporal resolution less
than 50 kyrs (Wu et al., 2014). Only mudstone samples were selected and analyzed for paleoclimatic
and paleoenvironmental inferences, as based on our previous work clay minerals in siltstones and
sandstones may be formed by authigenesis during early diagenetic process (Gao et al., 2013; 2015b),
as such, the previously generated datasets were also filtered for mudstone samples only.

Clay mineralogy was determined by X-ray diffraction (XRD) analysis. Bulk rock samples were

slightly ground and reacted with 0.1 N HCl to remove carbonates. They were then deflocculated by
successive washing with distilled water, and particles smaller than 2 μm were separated by



sedimentation and centrifugation (Liu et al., 2004). Clay-sized minerals (<2 μm) were analyzed using
XRD on oriented mounts of non-calcareous clay-sized particles (Liu et al., 2004). XRD was carried
out on a PANalytical X'Pert PRO diffractometer with Cu Kα radiation and Ni filter, under 40 kV
voltage and 25 mA intensity, at the State Key Laboratory of Marine Geology, Tongji University.
Three XRD runs were performed on each sample, following air-drying, ethylene-glycol solvation for
24 hours, and heating at 490 ℃ for 2 hours. The goniometer performed a scan from 3˚ to 30˚ 2Θ for
each run.

Identification of clay minerals was based on the positions of the (001) basal reflections on the

XRD diffractograms under the three different conditions (Moore and Reynolds, 1997; Figure 3). In the
present study, smectite includes randomly ordered mixed-layer illite-smectite, with a diagnostic
expanded 17 Å peak upon ethylene-glycol treatment. Semi-quantitative calculations were carried out
on the XRD patterns under ethylene glycol-solution conditions, using the MacDiff software (Petschick
et al., 1996). The relative abundances of each clay-mineral species were estimated mainly according to
the areas of the (001) series of basal reflections, i.e. smectites 17 Å, illite 10 Å, and kaolinite/chlorite 7
Å (Liu et al., 2004; Figure 3). Relative proportions of kaolinite and chlorite were determined using the
ratios of the 3.57/3.53 Å peak areas (Liu et al., 2004). Ratios of phyllosilicate clay minerals and quartz
in clay fractions (clay/quartz ratio) were determined by ratios between a sum of clay peak (17 Å + 10
Å + 7 Å) areas and the quartz 4.26 Å peak area (Frank and Ehrmann, 2010).

The illite chemistry index was applied to estimate intensity of chemical weathering in the present

study. It is based on the area ratio of the 5 Å peak and the 10 Å peak under ethylene glycol-solution
conditions (Liu et al., 2012; Figure 3). Values of this index above 0.40 represent Al-rich illites
(muscovites) which are products of strong hydrolysis. When Mg and Fe substitute Al in illite's crystal
lattice, this index decreases accordingly. Ratios below 0.15 are found in Fe–Mg-rich illites (biotites),
which are characteristic of physical erosion (Petschick et al., 1996). Smectite and illite crystallinity
indexes, which are half-height widths of 17 Å peak and 10 Å peak under ethylene glycol conditions,
are positively correlated to weight percentages of smectite and illite respectively, which are calculated





on peak areas (see Supplement). Therefore, given the purpose of this study, we use proxies based on
the ratios of different peak areas (e.g., clay/quartz ratio, illite chemistry index) as paleoclimatic and
paleoenvironmental indicators in the present study, rather than measures of the peak shape such as
smectite and illite crystallinity index.

**4. Results**

Our results indicate that clay minerals in the SMF of the SK-1 cores are dominated by smectite

(1-99%) and illite (1-92%), with average weight percentages of 68% and 26% respectively (Figure 2).
Kaolinite (0-12%) and chlorite (0-13%) are minor clay species with average abundances of 3% and 4%
respectively (Figure 2). Overall smectite shows an increasing trend in relative proportion from bottom
to top of the SMF, whereas illite, kaolinite and chlorite show corresponding decreasing trends, which
results in an increasing trend of smectite/illite ratio over the SMF. Illite chemistry index increases
gradually with decreasing depth. The clay/quartz ratio varies between 0-200 and has higher values in
the upper part of the SMF (Figure 2).

In addition to these overall trends, short-term fluctuations on relative proportions and ratios of

clay species and illite chemistry index are observed at approximately the hundred-meter scale (Figure
2). Eight zones are divided according to synchronous or inverse changes among proxies. For example,
zones II, V, VIII are characterized by high smectite proportion and low illite proportion (high
smectite/illite ratio), high illite chemistry index and high clay/quartz ratio (Figure 2). On the contrary,
lower smectite content and higher illite content (lower smectite/illite ratio), lower illite chemistry
index and lower clay/quartz ratio are observed in zones I, IV, VI (Figure 2). These features
demonstrating the co-evolution of mineralogical composition and crystallinity imply a unified
controlling mechanism. Furthermore, zones III and VII show different features from other clay
mineralogical zones. Zone III has generally high smectite content and low illite content, but with
lower illite chemistry index and clay/quartz ratio, except for the one peak of extremely low
smectite/illite ratio in the middle of the zone (Figure 2). Zone VII is characterized by a moderate
smectite/illite ratio but increasing values of illite chemistry index and an increase in kaolinite content
in clay mineralogical composition (Figure 2).

**5. Discussion**
**5.1 Origin and paleoclimatic significance of clay minerals in the SMF of the SK-1n core**

In sedimentary basins, clay mineralogical composition in mudstones are controlled by several

factors, including the weathering of parent rocks, differential settling in transportation and deposition
processes, pedogenic transformation and neoformation in paleosols, and diagenesis (Chamley, 1989;
Moore and Reynolds, 1997; Wilson, 1999; Thiry, 2000; Gao et al., 2015b; Deconinck et al., 2019). It
is therefore important to ensure that clay minerals are primarily detrital in origin without significant
influence of diagenesis, before they are used for paleoenvironmental reconstructions.

Gao et al. (2015b) examined clay minerals in all geological formations of the SK-1 cores at a

sampling interval of ~10 m, and suggested that burial diagenesis could cause the decreasing trend of
smectite, increasing trend of illite, and ordered smectite-illite mixed layers and chlorite with depth
from ~1000 m through ~2000 m. However, given the absence of ordered smectite-illite mixed layers
and oscillating depth-dependent variations of smectite and illite, burial diagenesis appears to be
negligibly influencing clay minerals of the SMF at depths shallower than 1000 m (Gao et al., 2015b).
A high content of smectite in rose-like shape was detected in sandstones of the SMF, likely as a result
of authigenesis during early diagenesis (Gao et al., 2013; 2015b). On the contrary, in the mudstone
unites the dominance of smectite and illite, and their platy shapes under electron microscope, indicate
a detrital origin likely linked to changes in weathering regime and paleoclimatic changes (Gao et al.,
2015b).



The most abundant clay mineral in the SMF is smectite, in which randomly ordered smectite-illite mixed layers are included because of their similar origin and paleoclimatic significance (Figures 2 and 3). Two main origins of sedimentary smectite are chemical weathering of volcanic rocks and transformation and neoformation during pedogenesis in soil profiles (Deconinck and Chamley, 1995; Wilson, 1999; Liu et al., 2009). During the Campanian and Maastrichtian, the main sources of sediments in the Songliao Basin were the Zhangguangcai Range and the Lesser Xing'an Range to the east of the basin, in response to uplift caused by subduction of the Pacific plate (Feng et al., 2010; Zhang et al., 2017; Figure 1B). Today these mountain ranges primarily expose granitic rocks, but a large suite of geochemical provenance data indicates that during the Cretaceous period mafic volcanic rocks were present and provided sedimentary sources (Gao et al., 2013; Xu et al., 2013; Meng et al., 2014). Thus, the weathering of volcanic rocks in the Lesser Xing'an – Zhangguangcai ranges could be a potential source for smectite in SMF of the Songliao Basin. Reworking or in-situ formation of smectitic soils may be another source. Smectite tends to form in soils of low-lying topography, poor drainage and base-rich parent material, such as Vertisols and Alfisols, through the neoformation or transformation by mica minerals (Wilson, 1999). It has been reported that multiple layers of paleosols occurred in the SMF of the SK-1n core, whereas widespread floodplains across the basin could have favored paleosol development under a temperate, semi-humid to semi-arid climate in the latest Cretaceous (Huang et al., 2013; Wang et al., 2013; Gao et al., 2015a).

Illites are usually physical weathering products of crystalline rocks (Chamley, 1989). In soils illites are commonly inherited from parent rocks and do not typically form during pedogenesis (Wilson, 1999). We propose that the illites in the SMF of the SK-1n core are primarily derived from the physical weathering of granitoids in the Lesser Xing'an – Zhangguangcai ranges (Gao et al., 2013). The chemical index of illite, which represents chemical composition of illitic minerals, is therefore a useful indicator for weathering intensity in the source area, where higher values indicate stronger hydrolysis whereas lower values indicate stronger physical erosion (Petschick et al., 1996; Liu et al., 2012). Furthermore, as smectite fractions are usually finer and lighter than other clay minerals, they



tend to be transported further and deposited at distal lacustrine environment, whereas illite and
kaolinite could be preferentially deposited at proximal lacustrine and fluvial environments (Thiry,
2000). Differentially settling during sedimentary processes may also influence the composition of clay
minerals in mudstones such as those in the SMF.

Although we cannot fully distinguish the influences of parent rock weathering, pedogenic

formation and differential settling on origins and relative proportions of smectites and illites, we argue
that these factors drive the ratio of smectites and illites in the same direction during hydroclimate
changes. For example, in a wetter hydroclimate, with an intensified hydrologic cycle, increased
chemical weathering on parent rocks and higher rates of transformation and neoformation in soil
profiles are expected to generate more smectite versus illite. Expanded lakes may also preserve more
differentially deposited smectite in lacustrine sediments. On the contrary, drier hydroclimate
conditions would favor stronger physical weathering, decreased smectite formation in soils, and more
illite deposition in fluvial sediments. The ratio of smecitite/illite contents is therefore applied as a
paleoclimatic proxy, where higher ratios indicate more humid climate.

Kaolinite is commonly formed under stronger hydrolysis which is typical in tropical regions,

whereas chlorite is typically considered a clay species derived from physical weathering of crystalline
rocks (Chamley, 1989). Both minerals are minor in the SMF but kaolinite increases slightly in some
intervals, indicating increased hydrolysis (Figure 2). The ratio between clay minerals and clay-sized
quartz can be used as a paleoclimatic indicator because stronger chemical weathering under more
humid climate would produce more clays compared to quartz (Chamley, 1989).

To summarize our interpretation, clay minerals in the SMF are mainly originated from

weathering of parent rocks and/or pedogenesis and are useful for making inferences about terrestrial
climate changes in the Songliao Basin. As such, we utilize three paleoclimatic proxy timeseries,
sensitive to hydroclimate change, from our clay mineralogical records, the smectite/illite ratio, the
illite chemistry index and the clay/quartz ratio, where higher (lower) values of these proxies indicate
stronger (weaker) chemical weathering conditions and more (less) humid climate.



### 5.2 Terrestrial paleoclimate evolution of the Songliao Basin in the latest Cretaceous

### inferred from clay mineralogical proxies

During the last ten million years of the Cretaceous Period, records derived from marine sediments suggest an overall cooling trend of global climate that was punctuated by several short-term cooling and warming events (Barrera and Savin, 1999; Friedrich et al., 2012; O'brien et al., 2017). However, very few terrestrial records on these short-term (sub-Myr) climatic events in the latest Cretaceous have been reported (Nordt et al., 2003; Salazar-Jaramillo et al., 2016), due to both difficulties in age control and the discontinuous nature of the terrestrial sedimentary record. The SMF of the SK-1n core in the Songliao Basin provides one of the best-preserved terrestrial records spanning latest Cretaceous through earliest Paleogene in the world (Wang et al., 2013; Gao et al., 2015a; Zhang et al., 2018). Previous stable isotopic and paleontological studies indicate paleoclimate changes are consistent with global trends (Gao et al, 1999; Wang et al., 2013; Gao et al., 2015a; Zhang et al., 2018). In the following section we discuss clay mineralogical evidences for changes of terrestrial climate over the entire interval and around the K-Pg boundary, especially as they relate to hydroclimate and weathering intensity changes, and correlate these observations with regional and global records (Figures 4 and 5).

The most prominent cooling event in the latest Cretaceous occurred at ~71-70 Ma, when oxygen isotopes of benthic foraminifera increased by about 1‰ (Barrera and Savin, 1999; Miller et al., 2005; Friedrich et al., 2012; Figure 4). Although two different processes, buildup of Antarctic glaciation and invasion of high-latitude cold water to tropical and subtropical oceans, have been used to interpret these isotopic excursions (Miller et al., 2005; Jung et al., 2013), there is a consensus that global climate cooled in the early Maastrichtian, which is also supported by sedimentary and palynological evidences for sea ice in polar regions (Davies et al., 2009; Bowman et al., 2013). In the Songliao Basin a striking negative excursion of oxygen isotopes in pedogenic carbonates and a contemporaneous positive excursion of carbon isotopes can be observed at ~70.5 Ma, which are interpreted as responses to terrestrial cooling and/or drying with more westerly-sourced precipitation



(Gao et al., 2015a). Our clay mineralogical records in the SMF of SK-1n core show that smectite/illite
ratio, illite chemistry index and clay/quartz ratio all have lower values during this time interval (Zone
IV in Figures 2 and 4). These indicate drier climate and stronger physical weathering that favor
fragmentation of parent rocks and generation of illitic clay minerals. Climate cooling in early
Maastrichtian may have strengthened the westerlies in northern mid-latitude regions and weakened the
Pacific-sourced air masses, which would have resulted in a more arid condition over the Songliao
Basin (Gao et al., 2015a) and its provenance regions (i.e., basin-wide weathering zones forming clay
minerals). Similar mechanisms may have controlled the cooler and drier terrestrial climate from ~68.5
Ma to ~66.5 Ma, when marine oxygen isotopes were as high as ~0.5-1.0 ‰ (Barrera and Savin, 1999;
Jung et al., 2013; Figure 4). All clay mineralogical indexes have decreasing trends during this 2-myr
time period, indicating a more arid climate and stronger physical weathering (Zone VI in Figure 4).

Global climatic warming events occurred at ~69.5-68.5 Ma and ~66.4-66.1 Ma, both supported

by negative excursions of oxygen isotopes in benthic foraminifera (Barrera and Savin, 1999; Jung et
al., 2013; Figure 4), which are also known as the Mid-Maastrichtian Event (MME) and Late
Maastrichtian Event (LME). The LME will be further discussed in the following section as its
potential linkage to Deccan Traps volcanism and the mass extinction at the K-Pg boundary (Hull et al.,
2020). The MME is characterized by increasing temperatures and perturbations in the carbon cycle in
both marine and terrestrial realms, probably related to the Ninety East Ridge volcanism erupted ~69.5
million years ago in the Indian Ocean (Nordt et al., 2003; Salazar-Jaramillo et al., 2016; Mateo et al.,
2017). A positive $\delta^{18}O$ excursion and a contemporaneous negative $\delta^{13}C$ excursion in pedogenic
carbonates in the Songliao Basin during MME are interpreted as increasing temperature, precipitation
and moisture delivery from Pacific, following an opposite mechanism to climate cooling (Gao et al.,
2015a). Higher values of smectite/illite ratio, illite chemistry index and clay/quartz ratio presented
here in the SMF indicate more humid climate and stronger chemical weathering (Zone V in Figures 2
and 4). Our clay mineralogical records further outline another potential warming period, ~74-72 Ma,
when illite chemistry index and clay/quartz ratio have higher values (Zone II in Figure 4).



Smectite/Illite ratio is elevated during ~74-72 Ma compared with that of ~76-74 Ma. A slight decrease
in marine $\delta^{18}O$ and pedogenic carbonate $\delta^{18}O$, and the increasing trend of pedogenic carbonate $\delta^{13}C$
further support climate warming and wetting, although not as strongly as during the MME (Figure 4).
It is noteworthy that two intervals do not follow the trends of clay mineralogical changes as
described above. During ~72-70.5 Ma, two sets of grey to black mudstones of semi-deep lacustrine
facies deposited in the lower part of the Mingshui Formation, which are separated by an interval of
grey sandstone and red mudstone of fluvial channel and floodplain facies (Zone III in Figures 2; Wang
et al., 2015). Episodically intensified extensional and compressional stress fields have been applied to
interpret the sudden changes in sedimentary environments. This is supported by evidence from
regional seismic analysis, paleontological data and the discovery of mafic dykes (Zhang et al., 2009;
Cheng et al., 2018). The lacustrine grey mudstones have higher smectite/illite ratio but lower
clay/quartz ratio, whereas the floodplain red mudstones have lower values of smectite/illite ratio, illite
chemistry index and clay/quartz ratio (Zone III in Figures 2 and 4). We tentatively interpret that
stronger tectonism induced physical weathering and therefore lead to higher illite and quartz
production, low illite chemistry index and low clay/quartz ratio, although lake expansions, as a result
of tectonic extension, may have caused preferential deposition of smectite versus illite in semi-deep
lacustrine environments (Figures 2 and 4; see also previous discussion).
During the time interval of ~76-74 Ma, warmer and drier climate in the Songliao Basin is
recorded by lower values of all clay proxies, high temperatures derived by clumped isotopes of
pedogenic carbonates, high $\delta^{13}C$ values in pedogenic carbonate and predominant dry taxa in the pollen
and spore assemblages (Zone I in Figure 4; Gao et al., 1999; Wang et al., 2013; Gao et al., 2015a;
Zhang et al., 2018). A contemporaneous decreasing trend in marine $\delta^{18}O$ seems to support a warmer
period (Jung et al., 2013). The reason for a warmer but drier climate state over the Songliao Basin in
the late Campanian is not clear yet. One possible explanation could be that high coastal mountains
along the eastern margin of the East Asia continent blocked Pacific moisture and caused rain shadow
effect in the Songliao Basin and other East China basins (Zhang et al., 2016). The elevation of the





coastal mountains could be reduced during the Maastrichtian, probably due to continuous weathering
and erosion, allowing Pacific moisture to invade into the Songliao Basin region during subsequent
warming intervals.
**5.3 Terrestrial paleoclimate changes in the Songliao Basin across the K-Pg boundary and**
**correlations with global records**
The massive extinction at K-Pg boundary is the last of the five largest Phanerozoic massive
extinction events (Raup and Sepkoski, 1982; Petersen et al., 2016; Hull et al., 2020). Debates remain
on causes of this mass extinction event, with the Chicxulub asteroid impact and the Deccan Trap
volcanism as two most cited candidates, both of which would have caused dramatic environmental
changes on earth (Schulte et al., 2010; Keller, 2014; Schoene et al., 2019; Sprain et al., 2019). A
recent study using carbon cycle modeling and global paleotemperature compilations supports the
hypothesis that outgassing of Deccan Trap volcanism caused climatic warming before and after the K-
Pg boundary, although the after-boundary warming is limited by extinction-related carbon cycle
perturbations (Hull et al., 2020).
The clay mineralogical and stable isotopic records of the SMF in Songliao Basin further support
climatic perturbations on land across the K-Pg boundary (Figure 5). Globally, the late Maastrichtian
warming event, featured by an elevation of ~2 to 4 °C in both marine and terrestrial temperatures
within ~300 thousand years, was followed by a transient cooling to pre-LME temperatures right
before the K-Pg boundary (Nordt et al., 2003; Wilf et al., 2003; Petersen et al., 2016; Barnet et al.,
2018; Woelders et al., 2018; Hull et al., 2020; Figure 5). In the Songliao Basin, clumped isotopes of
pedogenic carbonates indicate a carbonate formation temperature (likely summer soil temperature)
rise in LME and a drop after LME but before the K-Pg boundary (Zhang et al., 2018; Figure 5).
Besides, positive $\delta^{18}O$ and negative $\delta^{13}C$ excursions in pedogenic carbonates inside of the LME
support increasing temperature, humidity and delivery of Pacific moisture (Gao et al., 2015a; Figure
5). An increase in illite chemistry index together with a slight increase in clay/quartz ratio suggests
that a more humid climate and stronger chemical weathering was due to warming and wetting (pink





band of Zone VII in Figure 5). However, we observe no significant changes in smectite/illite ratios
during the LME. After the LME but before the K-Pg boundary, smectite/illite ratio, illite chemistry
index, clay/quartz ratio and land temperature all decrease likely as a response to the transient cooling
event (blue band of Zone VII in Figure 5).

In the earliest Danian, global temperatures gradually increased by >1 °C higher than the pre-

LME level in about 600 thousand years, probably due to combined effects of post-boundary $CO_2$
outgassing by Deccan Traps volcanism and extinction related carbon cycle perturbation (Hull et al.,
2020). In the Songliao Basin we observe rising summer temperature, increasing $\delta^{18}O$ but decreasing
$\delta^{13}C$ in pedogenic carbonate, increasing illite chemistry index, suggesting warmer and more humid
climate with more intensive chemical weathering (Zone VIII in Figure 5). Increases in smectite/illite
ratio and clay/quartz ratio are observed during this period, possibly because of a combination of
"lagged" clay formation caused by enhanced weathering during the LME (i.e., Deccan Volcanism)
and new formation during the post K-Pg warming period.

It is notable that there is an apparently dampened response to the LME and the K-Pg boundary in

our clay mineralogy proxies (Figure 5). We propose that this is probably due to the relatively long
response time of terrestrial weathering regimes to an enhanced hydrologic cycle and increased
temperatures (up to 500 kyrs; Walker et al., 1981; Archer et al., 2005). Given the immediate nature of
the K-Pg impact and the short duration of the preceding Deccan Traps volcanism (~200 kyrs; Barnet
et al., 2018; Schoene et al., 2019; Sprain et al., 2019), we do not expect weathering systems such as
the Songliao Basin to react temporally in sync with short term perturbations to the carbon cycle,
though clearly further work on other terrestrial sections is required to confirm this hypothesis.

**6. Conclusions**

High-resolution clay mineralogical analysis has been conducted on mudstones of the Sifangtai

Formation and the Mingshui Formation in the SK-1n scientific core of the Songliao Basin, NE China



to study terrestrial paleoclimatic changes from the latest Cretaceous through the earliest Paleogene.
The clay mineralogy is dominated by smectite and illite, with minor contributions from kaolinite and
chlorite. As these clay species originate from the weathering of parent rocks and/or paleosols, three
clay mineralogical indicators (smectite/illite ratio, illite chemistry index and clay/quartz ratio) were
used to reconstruct paleoclimate and paleoenvironment and were correlated with global paleoclimatic
records. During warming intervals from the latest Cretaceous through the earliest Paleogene, values of
smectite/illite ratio, illite chemistry index and clay/quartz ratio all increase, representing a more humid
climate and stronger chemical weathering. Opposite trends in these clay mineralogical proxies were
observed during cooling intervals, corresponding to less humid and weaker chemical weathering.
Across the Cretaceous-Paleogene boundary, climatic warming and cooling events related to Deccan
Traps volcanism and massive biologic extinction were recorded in the Songliao Basin by changes in
clay mineralogical composition, illite chemistry index and isotopic composition of pedogenic
carbonates, which indicate fluctuations in precipitation and weathering intensity. Our work
demonstrates that terrestrial hydroclimate and weathering regimes in the mid-latitude Songliao Basin
fluctuated during the latest Cretaceous through the earliest Paleogene and sensitively responded to
global climate changes.

**Acknowledgements**
This study was supported by the National Natural Science Foundation of China (41602116,
41790450, 41972096). D.E.I. is supported by Miller Research Institute and UC President's
Postdoctoral Fellowships.

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

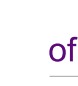
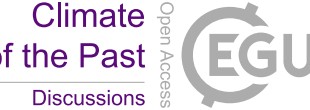

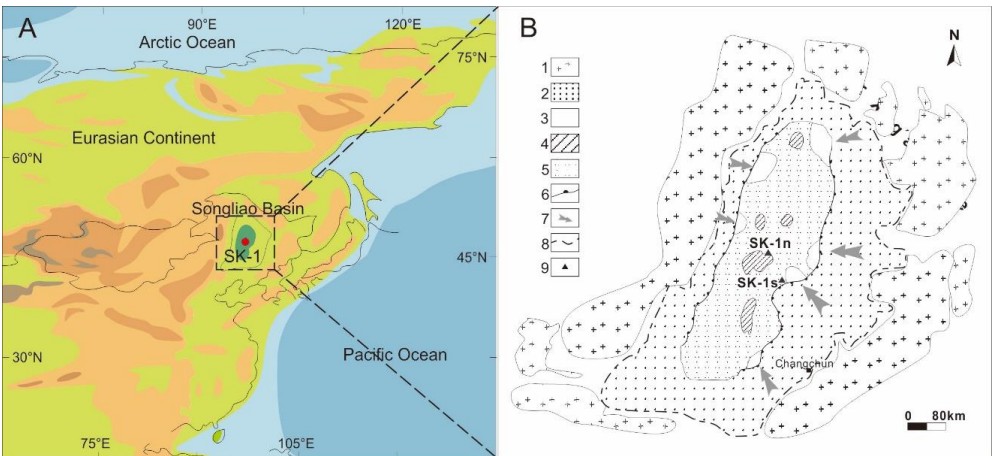

**Figure 1.** A: Paleogeographic setting of the Songliao Basin (dashed line) and the SK-1 scientific

drilling site (red dot) at ~70 Ma. Dark green area approximates depositional limits of SMF. Black

solid lines are approximate country boundaries. Modified after Gao et al. (2015a). B: Geological

setting on the Songliao Basin and the border mountain ranges during the deposition of SMF, which

shows sedimentary environments, provenance directions and drilling sites of the SK-1 boreholes.

Labels are: 1—Phanerozoic granitoids; 2—Sediments deposited before Mingshui Formation; 3—

Alluvial fan deposits; 4—Lacustrine deposits; 5—Alluvial plain deposits; 6—Erosion

boundary at Mingshui Formation deposition time; 7—Provenance and sediment transportation

direction; 8—Basin boundary; 9—SK-1n and SK-1s drilling sites. Modified from Wu et al. (2011) and

Gao et al. (2013).



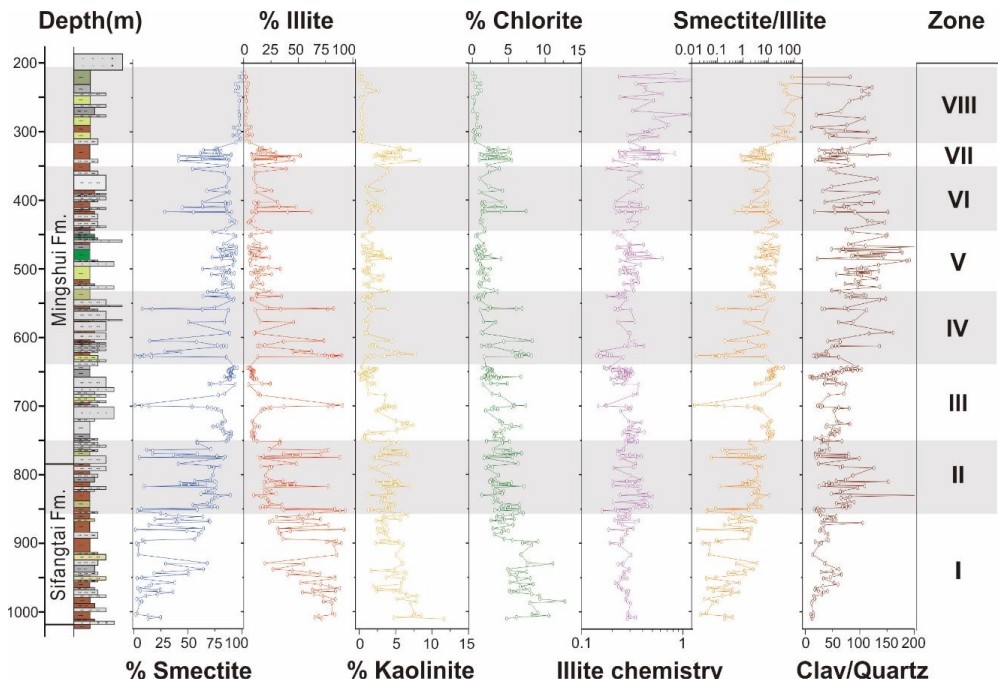

**Figure 2.** Clay mineralogical indexes in the SMF of the SK-1n core. Illite chemistry – illite chemistry index, Smectite/Illite – ratios of relative proportions between smectite and illite, Clay/Quartz – ratios of relative proportions between phyllosilicate clay minerals and quartz in clay fractions.





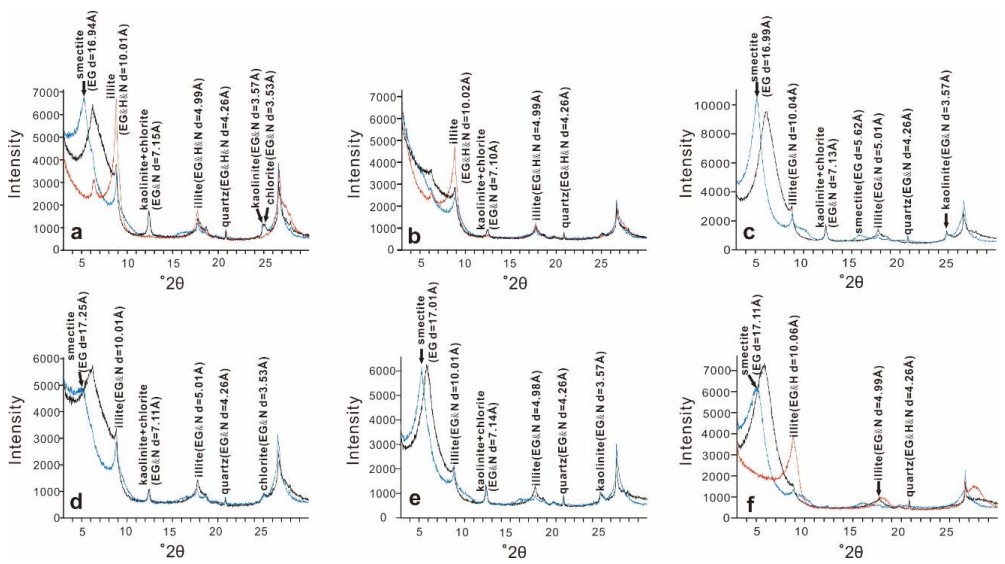

687

**Figure 3.** Typical X-ray diffraction diagrams of the SMF sediments in the SK-1n core. (a) green

mudstone at 801.0 m (smectite 73%, illite 19%, kaolinite 4%, chlorite 4%). (b) brown mudstone at

558.0 m (smectite 8%, illite 82%, kaolinite 4%, chlorite 7%). (c) brown mudstone at 494.0 m

(smectite 92%, illite 5%, kaolinite 2%, chlorite 1%). (d) brown mudstone at 409.0 m (smectite 64%,

illite 32%, kaolinite 2%, chlorite 3%). (e) brown mudstone at 341.0 m (smectite 88%, illite 7%,

kaolinite 3%, chlorite 1%). (f) green mudstone at 298.0 m (smectite 97%, illite 3%, kaolinite 0%,

chlorite 0%).

695

**Climate**
**of the Past**
Discussions
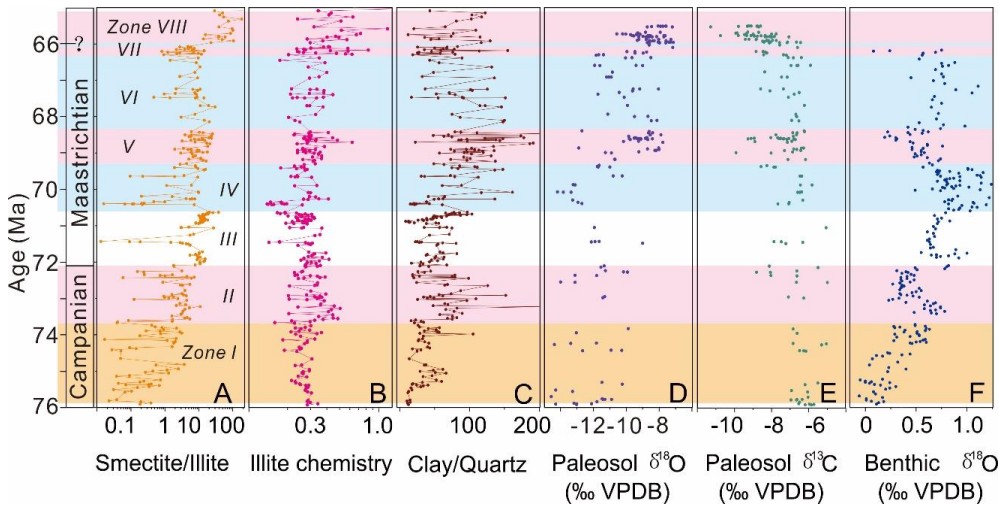

696

**Figure 4.** Latest Cretaceous terrestrial paleoclimatic records of the SMF in the Songliao Basin and

correlations to marine records. A-C. clay mineralogical indicators of paleoclimate in the SMF. Zones I

to VIII refer to clay mineralogical zones in Figure 2. D-E. compiled stable oxygen and carbon isotopes

of pedogenic carbonates in the SMF (data sources are Huang et al., 2013; Gao et al., 2015a; Zhang et

al., 2018). F. stable oxygen isotopes of benthic foraminifera in Oceanic Drilling Program site 1210,

central Pacific (Jung et al., 2013). Pink, blue and yellow bands indicate warmer-wetter, cooler-drier

and warmer-drier climate intervals respectively.

704





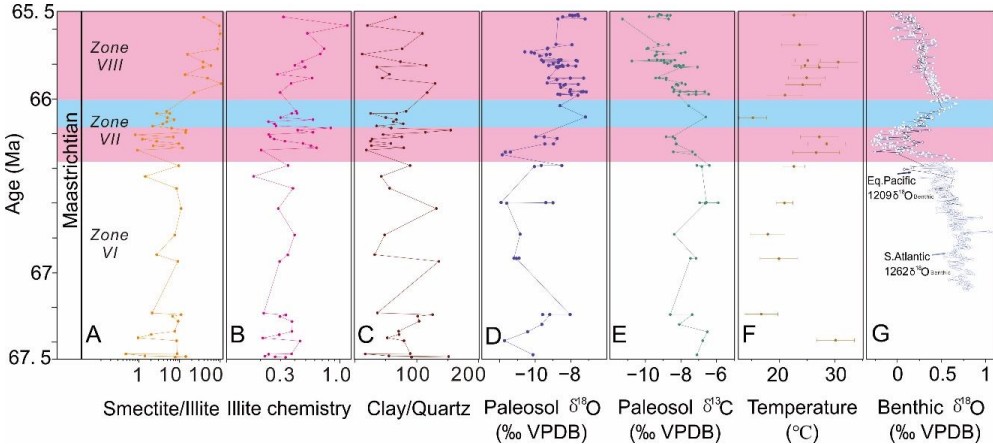

**Figure 5.** Terrestrial paleoclimatic records of the Songliao Basin and correlation with marine records

across the Cretaceous-Paleogene Boundary. A-D. clay mineralogical indicators of paleoclimate in the

Songliao Basin. See Figure 2 for abbreviations. E-F. compiled stable oxygen and carbon isotopes of

pedogenic carbonates in the SMF (data sources are Huang et al., 2013; Gao et al., 2015a; Zhang et al.,

2018). G. formation temperature of pedogenic carbonate in the Songliao Basin (Zhang et al., 2018). H.

stable oxygen isotopes of benthic foraminifera in Oceanic Drilling Program sites 1209 (central Pacific)

and site 1262 (south Atlantic) (data from Barnet et al., 2018). Pink and blue bands indicate warmer-

wetter and cooler-drier climate intervals respectively.