# Peer review of "Clay mineralogical evidence for mid-latitude terrestrial climate change from the"

_Climate of the Past, 2020_

## Short Comment (SC1) · 3 Apr 2020

This manuscript reported the result from the SK-1n core. Authors tried to interpret the paleoclimate change using the clay mineral information. This is very important since the numerous data and the interprets from the oceanic sediments, but the study from the continent is very rare. The constraints are the difficult to obtain the complete and continuous samples. The SK-1n carried in a long-lived lake across late cretaceous to paleogene and satisfied the study of paleoclimate. It is of course very important and is consistent to the air of the journal. I recommend the accept of this maunclipt.

[Figure]

I have some suggestion for this manuscript, the detail is following.

The methodology is right, but the reviewer is still worrying. The clay mineral was separated from the mudstone without the effect from diagenetic. This also means that the clay mineral was formed on continent, but we have not the idea about the weathering durations. I think authors might need to clarify how theses clay mineral can record the response from climate? It is nor like benthic foraminifera doing.

The description of results is too simple and fail to give the necessary details. Such as the detail of the zone I to VIII. And the overall trends need to be clarified systematically.

Line 202: crystallinity index of smectite and illite was calculated from a FWHM, the relation of CI and contents of mineral show a positive correlation, this might be misunderstood. The lower content of the clay mineral, the peak will be very weak and FWHM might be abnormal wider than the higher content ones.

Though the origin of the parent rock was deducted, it is too simple more support might be summarized from the published references. Line 261 "mafic volcanic"was mentioned, how the weathering of mafic volcanic produce smectite and illite? Illite was referred to the product of the physical weathering, but some reference suggested that illite was the strong chemical weathering of muscovite (muscovite was thought to be chemical stable mineral and widely spread in sandtone and mudstone)? Hence, the authors might think again about the physical weathering and chemical weathering of the clay minerals. Line 417-419, mentioned the chemical weathering origin of the illite.

Then, the trends from clay mineral might be delayed to some extents since its weathering from parent rock on the continental. Do authors find some abnormal trend might be affected by clay minerals?

The conclusion remark is not well documented. Lines 432-439 may be deleted since it is the repetition of the results.

---

## Referee Comment (RC2) · Anonymous Referee #2 · 9 Apr 2020

My general comments on Gao et al. "Clay mineralogical evidence for mid-latitude terrestrial climate change from the latest Cretaceous through the earliest Paleogene in the Songliao Basin, NE China" are summarized here. The Earth's climate state of late Cretaceous was characterized by high atmospheric $CO_2$ and global warmth. Significantly, the late Cretaceous era is a key period for biotic evolution. However, our knowledge of Earth's state during this period is mostly from marine records, owing to a lack of well-dated high-resolution terrestrial records. The paper by Gao et al. presents well-dated high-resolution clay mineralogical records of the Upper Cretaceous and Lower

Paleocene terrestrial deposition from the Songliao Basin, northeastern China. Their data shown that the relative percentages of the clay minerals were mainly controlled by regional paleoclimate and sedimentary environment, and they utilized three clay mineralogical proxies for paleoclimatic reconstruction. They found these proxies varied sensitively in responed to global climate changes during the latest Cretaceous to the earliest Paleogene. Their clay mineral data provide independent evidence for climate changes through the latest Cretaceous to the earliest Paleogene from the Songliao Basin, and would shed light on further investigations in biotic evolutionary during this period. The layout of this paper is good and the writing is in good shape. As such, I think this paper deserves to be published in Climate of the Past and potentially to be interested by a wide range of readers, albeit some aspects are still need to be improved. The following are some weak points that in my opinion should be addressed to improve the paper.

My main issue with the study is that the authors have to further justify their clay mineralogical proxies are reliable proxies for paleoclimatic reconstruction. In fact, in addition to climate and weathering, many other factors, such as provenance, recycling of sedimentary parent rocks, transport processes, and depositional environments, all could influence the type and proportion of clay minerals. I suggest the authors to evaluate to what extent these factors have influenced their clay mineralogical proxies.

Lines 24-26: These clay minerals can be also sourced from recycled sedimentary parent rocks. How to preclude this? See the comment above.

Lines 97-106: What are the new contributions of this paper beyond Gao et al. (2013; 2015b)?

Lines 106-108: I suggest the authors to mark the published data in their figs. 2, 4, and 5.

Lines 423-429: If the clay mineral proxies can only response to the long-duration climate events (>200 kyrs), then how to explain such many minima in their clay mineralogy proxies (such as percent of smectite, illite, and ratio of smectite/illite) in the figs. 2, 4, and 5?

Fig. 2: I suggest to 1) add the published observed magnetozones to geomagnetic polarity timescale (GPTS), 2) legend for lithostratigraphy, and 3) include labels like a), b), c) for the subplots.

———————————————————

---

## Referee Comment (RC3) · Anonymous Referee #3 · 9 May 2020

Dear Natascha Töpfer

Thanks for your invitation for the review of the manuscript submitted by Gao et al.(2020, doi:10.5194/cp-2020-36). The manuscript reported a clay mineralogical record from the late Cretaceous to early Paleocene, especially across the dramatic K-Pg boundary with a high-resolution record, and concluded a novel climate change based on clay mineralogical proxies. The paper is generally well-written and organized. The scope of this manuscript is well-chosen and will meet the broad interest for geologist. I give moderate revision because I think some issues which are the base of interpretations

need to be more discussed. The concerns are listed as follow:

1. The authors chose mudstones rather than sandstones to avoid the authigenesis during the diagenetic stage. However, how to prevent the influence of clay minerals from pedogenesis, which was widely developed throughout the SMF (in lines 152-153). In the pedogenic process, in general, clay minerals could form in solutions or transfer from other clays. The authors claim the clay minerals need to be primarily detrital in origin to use for paleoenvironmental reconstruction (Lines 240-241). However, two sources of smectite are discussed in the manuscript, in which in-situ formation is not excluded (Lines 264-265). It seems too simple to describe the factors affecting the interpretation of clay mineralogy, such as parent rock weathering, pedogenic formation, and differential settling on origins. (Lines 283-292). 2. According to Lines 188-194, the authors seem to add random mixed-layer illite-smectite to smectite when semi-quantifying the abundance of smectite based on the 17 Å peak area. As a matter of fact, the random mixed-layer illite-smectite could be a very wide peak between 10-15.5 Å under air-dry XRD pattern, which will split into two peaks at ∼17 Å and 10 Å after ethylene-glycol solvation. From Figure 3, we can tell the intensities of peaks at 10 Å enhanced after ethylene-glycol solvation besides the enhancements of peaks at ∼17 Å. From this point of view, the semi-quantitative amount of smectite could be questionable. Furthermore, the mixed-layer illite-smectite is an independent mineral phase, which could be the detrital phase from old strata or authigenic phase transformed from illite during pedogenesis. The amount of mixed-layer illite-smectite will likely affect the proportions of other clay minerals. Why the randomly ordered mixed-layer illtie-smectite and smectite have similar origin and paleoclimatic significance (Lines 253-254)? 3. In Figure 2, I suggest the authors add the age constrains, and then readers can know clay mineral trends and mutations along with the age. 4. I suggest the authors point out which pattern denotes what kind of treated slides in Figure 3. I can understand the black, blue, and red curves denote patterns of air-dry, ethylene-glycol solvation, and heating at 490 °C, respectively, which could not be the case for non-clay mineralogists. From the patterns of heating at 490 °C (Figure 3a, b, and f), we can read the peak at ∼14 Å

could be chlorite. However, why the authors did not present patterns of heating at 490 °C to further confirm having or having not chlorite in samples in Figure 3c, d, and e. 5. The author claimed the stronger chemical weathering under more humid climate would produce more clays compared to quartz (Line 297-298). I think it is promising on condition that it happened in in-situ pedogenic profile. However, the grain-size distribution in this study could largely depend on sedimentary process.

---

## Author Comment (AC1) · 10 Jun 2020

This manuscript reported the result from the SK-1n core. Authors tried to interpret the paleoclimate change using the clay mineral information. This is very important since the numerous data and the interprets from the oceanic sediments, but the study from the continent is very rare. The constraints are the difficult to obtain the complete and continuous samples. The SK-1n carried in a long-lived lake across late cretaceous to paleogene and satisfied the study of paleoclimate. It is of course very important and is consistent to the air of the journal. I recommend the accept of this mauncript.

Response: We appreciate the helpful comments by Anonymous Referee #1.

I have some suggestion for this manuscript, the detail is following. The methodology is right, but the reviewer is still worrying. The clay mineral was separated from the mudstone without the effect from diagenetic. This also means that the clay mineral was formed on continent, but we have not the idea about the weathering durations. I think authors might need to clarify how theses clay mineral can record the response from climate? It is nor like benthic foraminifera doing.

Response: We consider weathering of parent rocks and pedogenesis as two main origins of clay minerals of SMF. In a wetter hydroclimate, with an intensified hydrologic cycle, increased chemical weathering on parent rocks and higher rates of transformation and neoformation in soil profiles are expected to generate more smectite versus illite. We estimate >100 kyr for in-situ clay formation in soil profiles and several hundred thousand years for clay accumulation in sedimentary basins. We rewrote section "5.1 Origin and paleoclimatic significance of clay minerals in the SMF of the SK-1n core" to clarify these questions.

The description of results is too simple and fail to give the necessary details. Such as the detail of the zone I to VIII. And the overall trends need to be clarified systematically.

Response: We expanded the "Results" section following this comment.

Line 202: crystallinity index of smectite and illite was calculated from a FWHM, the relation of CI and contents of mineral show a positive correlation, this might be misunderstood. The lower content of the clay mineral, the peak will be very weak and FWHM might be abnormal wider than the higher content ones.

Response: We agree that weak peaks in XRD diffractograms may cause abnormally wider FWHM. This actually interprets the scattering of datapoints in illite crystallinity at lower illite percentages (see plot in supplementary file). However, the peak areas of high-percentage smectite or illite samples are much larger than those of lowpercentage, which cause much wider FWHM in high-percentage samples. We therefore consider this to be the reason for a generally positive correlation between percentage and crystallinity, and did not use crystallinity as a proxy for paleoclimate.

Though the origin of the parent rock was deducted, it is too simple more support might be summarized from the published references. Line 261 "mafic volcanic"was mentioned, how the weathering of mafic volcanic produce smectite and illite? Illite was referred to the product of the physical weathering, but some reference suggested that illite was the strong chemical weathering of muscovite (muscovite was thought to be chemical stable mineral and widely spread in sandtone and mudstone)? Hence, the authors might think again about the physical weathering and chemical weathering of the clay minerals. Line 417-419, mentioned the chemical weathering origin of the illite.

Response: We rewrote section "5.1 Origin and paleoclimatic significance of clay minerals in the SMF of the SK-1n core" to better constrain the origins of clay minerals and the rationales of clay mineralogical indicators as paleoclimatic proxies. In the provenance area of the SMF during the latest Cretaceous, both volcanic and granitic rocks were present. Illites of the SMF were primarily derived from the physical weathering and/or weak chemical weathering of granitoids in the Lesser Xing'an – Zhangguangcai ranges rocks. In lines 417-419 of original manuscript, we were not saying "chemical weathering origin of the illite". Furthermore, we consider increasing illite chemistry index, but not illite content, suggests warmer and more humid climate with more intensive chemical weathering during the LME warming event.

Then, the trends from clay mineral might be delayed to some extents since its weathering from parent rock on the continental. Do authors find some abnormal trend might be affected by clay minerals?

Response: We added more discussion on the response time of clay formation to climate change in the last paragraph of section "5.1 Origin and paleoclimatic significance of clay minerals in the SMF of the SK-1n core". In-situ formation of clay minerals in soil

profiles may take 50-100 kyr under a mid-latitude, temperate climate. Several hundred thousand years may be needed for clay accumulation in sedimentary basins through transportation and deposition of weathering products. Therefore, clay mineralogical proxies are reliable to reflect climatic changes at million-year timescales, but may not respond well to climatic events at shorter duration.

The conclusion remark is not well documented. Lines 432-439 may be deleted since it is the repetition of the results.

Response: We revised the "Conclusions" section.

---

## Author Comment (AC3) · 10 Jun 2020

My general comments on Gao et al. "Clay mineralogical evidence for mid-latitude terrestrial climate change from the latest Cretaceous through the earliest Paleogene in the Songliao Basin, NE China" are summarized here. The Earth's climate state of late Cretaceous was characterized by high atmospheric $CO_2$ and global warmth. Significantly, the late Cretaceous era is a key period for biotic evolution. However, our knowledge of Earth's state during this period is mostly from marine records, owing to a lack of well-dated high-resolution terrestrial records. The paper by Gao et al. presents welldated high-resolution clay mineralogical records of the Upper Cretaceous and Lower Paleocene terrestrial deposition from the Songliao Basin, northeastern China. Their data shown that the relative percentages of the clay minerals were mainly controlled by regional paleoclimate and sedimentary environment, and they utilized three clay mineralogical proxies for paleoclimatic reconstruction. They found these proxies varied sensitively in responded to global climate changes during the latest Cretaceous to the earliest Paleogene. Their clay mineral data provide independent evidence for climate changes through the latest Cretaceous to the earliest Paleogene from the Songliao Basin, and would shed light on further investigations in biotic evolutionary during this period. The layout of this paper is good and the writing is in good shape. As such, I think this paper deserves to be published in Climate of the Past and potentially to be interested by a wide range of readers, albeit some aspects are still need to be improved. The following are some weak points that in my opinion should be addressed to improve the paper.

Response: We appreciate the helpful comments by Anonymous Referee #2.

My main issue with the study is that the authors have to further justify their clay mineralogical proxies are reliable proxies for paleoclimatic reconstruction. In fact, in addition to climate and weathering, many other factors, such as provenance, recycling of sedimentary parent rocks, transport processes, and depositional environments, all could influence the type and proportion of clay minerals. I suggest the authors to evaluate to what extent these factors have influenced their clay mineralogical proxies. Lines 24-26: These clay minerals can be also sourced from recycled sedimentary parent rocks. How to preclude this? See the comment above.

Response: We rewrote section "5.1 Origin and paleoclimatic significance of clay minerals in the SMF of the SK-1n core" to better constrain the origins of clay minerals and the rationales of clay mineralogical indicators as paleoclimatic proxies. We consider weathering of parent rocks and pedogenesis as two main origins of clay minerals of SMF. In a wetter hydroclimate, with an intensified hydrologic cycle, increased chemical weathering on parent rocks and higher rates of transformation and neoformation in soil profiles are expected to generate more smectite versus illite, higher illite chemistry index, and more clay minerals versus clay-sized quartz. We also consider other sedimentary processes, such as differentially settling and sedimentary recycling, have little influence on our clay mineralogical records, because the Songliao Basin contained only small ponds or lakes and had a relatively flat morphology at the latest Cretaceous. Please see new text for more details.

Lines 97-106: What are the new contributions of this paper beyond Gao et al. (2013; 2015b)?

Response: We add 213 new data in the current paper, and include 91 data points published previously in Gao et al. (2013) and Gao et al. (2015b).

Lines 106-108: I suggest the authors to mark the published data in their figs. 2, 4, and 5.

Response: We marked new and published data with different symbols in Figure 2. However, we keep the same symbol for all data in Figures 4 and 5 so as not to distract readers, because these two figures are focused on terrestrial climatic evolution in the Songliao Basin and correlations to the paleosol stable isotope and global records.

Lines 423-429: If the clay mineral proxies can only response to the long-duration climate events (>200 kyrs), then how to explain such many minima in their clay mineralogy proxies (such as percent of smectite, illite, and ratio of smectite/illite) in the figs. 2, 4, and 5?

Response: We consider the minima and/or maxima in the clay mineralogy proxies actually reflect influences of in-situ clay formation during pedogenesis. In a paleosol profile, the clay composition in the top horizon and the bottom horizon may have differences larger than 50% (Maher et al., 2009).

Fig. 2: I suggest to 1) add the published observed magnetozones to geomagnetic

polarity timescale (GPTS), 2) legend for lithostratigraphy, and 3) include labels like a), b), c) for the subplots.

Response: We revised Figure 2 following this comment.

[Figure]

[Figure]

**Fig. 1.** revised figure 2

---

## Author Comment (AC4) · 10 Jun 2020

The manuscript reported a clay mineralogical record from the late Cretaceous to early Paleocene, especially across the dramatic K-Pg boundary with a high-resolution record, and concluded a novel climate change based on clay mineralogical proxies. The paper is generally well-written and organized. The scope of this manuscript is well-chosen and will meet the broad interest for geologist. I give moderate revision because I think some issues which are the base of interpretations need to be more discussed. The concerns are listed as follow:

Response: We appreciate the helpful comments by Anonymous Referee #3.

1. The authors chose mudstones rather than sandstones to avoid the authigenesis during the diagenetic stage. However, how to prevent the influence of clay minerals from pedogenesis, which was widely developed throughout the SMF (in lines 152-153). In the pedogenic process, in general, clay minerals could form in solutions or transfer from other clays. The authors claim the clay minerals need to be primarily detrital in origin to use for paleoenvironmental reconstruction (Lines 240-241). However, two sources of smectite are discussed in the manuscript, in which in-situ formation is not excluded (Lines 264-265). It seems too simple to describe the factors affecting the interpretation of clay mineralogy, such as parent rock weathering, pedogenic formation, and differential settling on origins. (Lines 283-292).

Response: We rewrote section "5.1 Origin and paleoclimatic significance of clay minerals in the SMF of the SK-1n core" to better constrain the origins of clay minerals and the rationales of clay mineralogical indicators as paleoclimatic proxies. We consider weathering of parent rocks and pedogenesis as two main origins of clay minerals of SMF. In a wetter hydroclimate, with an intensified hydrologic cycle, increased chemical weathering on parent rocks and higher rates of transformation and neoformation in soil profiles are expected to generate more smectite versus illite, higher illite chemistry index, and more clay minerals versus clay-sized quartz. We also consider other sedimentary processes, such as differentially settling and sedimentary recycling, have little influence on our clay mineralogical records, because the Songliao Basin contained only small ponds or lakes and had a relatively flat morphology at the latest Cretaceous. Please see text for more discussions.

2. According to Lines 188-194, the authors seem to add random mixed-layer illitesmectite to smectite when semi-quantifying the abundance of smectite based on the 17 Å peak area. As a matter of fact, the random mixed-layer illite-smectite could be a very wide peak between 10-15.5 Å under air-dry XRD pattern, which will split into two peaks at \_17 Å and 10 Å after ethylene-glycol solvation. From Figure 3, we can tell the CPD
intensities of peaks at 10 Å enhanced after ethylene-glycol solvation besides the enhancements of peaks at \_17 Å. From this point of view, the semi-quantitative amount of smectite could be questionable. Furthermore, the mixed-layer illite-smectite is an independent mineral phase, which could be the detrital phase from old strata or authigenic phase transformed from illite during pedogenesis. The amount of mixed-layer illite-smectite will likely affect the proportions of other clay minerals. Why the randomly ordered mixed-layer illite-smectite and smectite have similar origin and paleoclimatic significance (Lines 253-254)?

Response: Actually, none of the panels in Figure 3 show the intensities of peaks at 10 Å enhanced after ethylene-glycol solvation, but all the panels show enhancements of peaks at  $\sim$ 17 Å. This indicates smectite is the predominant mineral phase rather than I-S mixed layers. We consider the minor I-S mixed layers are also derived from weathering of feldspar, mica in parent rocks or transformation from illite during pedogeneis, similar to smectite.

3. In Figure 2, I suggest the authors add the age constrains, and then readers can know clay mineral trends and mutations along with the age.

Response: We revised Figure 2 following this comment.

4. I suggest the authors point out which pattern denotes what kind of treated slides in Figure 3. I can understand the black, blue, and red curves denote patterns of air-dry, ethylene-glycol solvation, and heating at 490 \_C, respectively, which could not be the case for non-clay mineralogists. From the patterns of heating at 490 \_C (Figure 3a, b, and f), we can read the peak at \_14 Å could be chlorite. However, why the authors did not present patterns of heating at 490\_C to further confirm having or having not chlorite in samples in Figure 3c, d, and e.

Response: We revised Figure 3 following this comment. If depth-adjacent samples have similar AD and EG curves, only selected samples were measured under heated condition. Therefore, patterns of heating were not present in Figure 3c, d, and e.

CPD
5. The author claimed the stronger chemical weathering under more humid climate would produce more clays compared to quartz (Line 297-298). I think it is promising on condition that it happened in in-situ pedogenic profile. However, the grain-size distribution in this study could largely depend on sedimentary process.

Response: We consider in both weathering profiles and soils, stronger chemical weathering under more humid climate would produce more clay minerals compared to claysized quartz, as the latter is more likely to form through physical fragmentation. Paleosol layers are very common throughout the SMF of the SK-1n core, which is further promising for the use of the clay/quartz ratio as a paleoclimatic proxy in the current study.